# How miRNAs Regulate Schwann Cells during Peripheral Nerve Regeneration—A Systemic Review

**DOI:** 10.3390/ijms23073440

**Published:** 2022-03-22

**Authors:** Anton Borger, Sarah Stadlmayr, Maximilian Haertinger, Lorenz Semmler, Paul Supper, Flavia Millesi, Christine Radtke

**Affiliations:** 1Department of Plastic, Reconstructive and Aesthetic Surgery, Medical University of Vienna, Währinger Gürtel 18-20, 1090 Vienna, Austria; anton.borger@meduniwien.ac.at (A.B.); sarah.stadlmayr@meduniwien.ac.at (S.S.); maximilian.haertinger@meduniwien.ac.at (M.H.); lorenz.semmler@meduniwien.ac.at (L.S.); paul.supper@meduniwien.ac.at (P.S.); flavia.millesi@meduniwien.ac.at (F.M.); 2Austrian Cluster for Tissue Regeneration, 1090 Vienna, Austria

**Keywords:** Schwann cell proliferation, Schwann cell migration, Wallerian degeneration, myelination, biomarker nerve injury, expression profile microRNA, pathway peripheral nerve, protein nerve, mRNA nerve, precision medicine

## Abstract

A growing body of studies indicate that small noncoding RNAs, especially microRNAs (miRNA), play a crucial role in response to peripheral nerve injuries. During Wallerian degeneration and regeneration processes, they orchestrate several pathways, in particular the MAPK, AKT, and EGR2 (KROX20) pathways. Certain miRNAs show specific expression profiles upon a nerve lesion correlating with the subsequent nerve regeneration stages such as dedifferentiation and with migration of Schwann cells, uptake of debris, neurite outgrowth and finally remyelination of regenerated axons. This review highlights (a) the specific expression profiles of miRNAs upon a nerve lesion and (b) how miRNAs regulate nerve regeneration by acting on distinct pathways and linked proteins. Shedding light on the role of miRNAs associated with peripheral nerve regeneration will help researchers to better understand the molecular mechanisms and deliver targets for precision medicine.

## 1. Introduction

Peripheral nerve injuries can be caused by trauma, tumor, compression, or by medical procedures [1]. High costs for treatment, rehabilitation, and occupational retraining are prevalent for the affected patients [2]. Nerve injuries can be divided into those with preserved integrity, called neuropraxia, and those with interrupted continuity, axonotmesis, and neurotmesis [3,4]. All types lead to an impediment of signal transduction inside the nerve, which results in a loss of sensory and motor function [3]. The first-line treatment option for neuropraxia is to eliminate any obstructions of nerve conduction by way of neurolysis. In case of axonotmesis or neurotmesis, the main goal is to restore continuity via end-to-end suture or via nerve grafts [5]. On cellular basis, in all types of axonotmesis, neurotmesis, and in severe cases of neuropraxia, so-called Wallerian Degeneration occurs [4,6].

### 1.1. Wallerian Degeneration

Wallerian Degeneration entails the entire disintegration of axons and the myelin sheath distal to the lesion site. Moreover, Schwann cells (SC) that sheath the axon and maintain hemostasis of the nerve environment transdifferentiate into proliferative and migrative repair types by halting the myelination process. Extracellular matrix and intraneural vessels become permeable to invading immune cells [7]. SCs, in cooperation with resident and invading macrophages, begin to clear the debris in the endoneurial tubes and pave the way for the outgrowing axon [7,8]. To facilitate and guide the outgrowing axons, SCs align and form Bands of Büngner, a guiding structure [8]. In the final phase, SCs redifferentiate into their adult phenotype, myelinating SCs resheath the axons, and the SCs maintain the integrative environment [9]. By the regeneration of axons through the lesion site, the connection between the spinal cord and the terminal organs is restored and the patient regains his motor and sensory functions.

However, a considerable share of patients demonstrates unsatisfying recovery after a surgical reconstruction [10,11,12]. Especially long-distanced and proximally defects as well as a delayed treatment are associated with poor outcome [10]. Underlying pathognomonic processes are a cellular growth arrest in long grafts [13] and a slow regeneration with a long denervation time of muscle, which results in a fatty atrophy of the muscle [4]. To overcome these issues, many approaches are investigated to increase the proliferation and migration of Schwann cells and to promote axonal outgrowth. The application of neurotrophic growth factors [14], cell-based therapies with mesenchymal stem cells (MSCs) [15], Schwann cells (SCs) [16], or extracellular vesicles (EVs) [17] are continuously explored to enhance the aforementioned crucial activities, which are associated with improved nerve regeneration.

### 1.2. miRNA

A promising approach for a precise modulation of beneficial processes are micro ribonucleic acids (miRNAs). miRNA are small noncoding RNAs regulating a large number of genes [18]. Previous publications already extensively addressed the biogenesis of miRNAs [19,20]. In brief, the precursor of a miRNA, named pri-miRNA, is transcribed in the nucleus by the RNA-Polymerase II from its respective DNA sequences. Subsequently, pri-miRNAs are processed into 65- to 70-nucleotide-long pre-miRNAs by a complex of Drosha and DiGeorge syndrome critical region 8 (DGCR8). Thereby, the pre-miRNA forms its characteristic hairpin structure. Under the participation of Exportin 5, the pre-miRNA is released into the cytosol [21], where the final maturation into 20- to 25-nucleotide-long miRNAs is executed by Dicer [20]. Matured miRNAs regulate gene expression by targeting mRNAs. In combination with the RNA-induced silencing complex (RISC), miRNAs either directly degrade mRNAs or block the translation through imperfect binding within the 3′-UTR binding region [22,23,24]. It is notable that, through imperfect binding, the same miRNA can regulate several mRNAs [25,26].

A standardized nomenclature for identified miRNAs is recommended [27]. The prefix gives the original species of a miRNA, for example, “hsa” for human (hsa-miR) origin and “rno” for rodents (rno-miR). The number in the suffix is a unique code for each microRNA. If there are minor variations at one or two positions in the nucleotide sequence, similar miRNAs receive a letter in the suffix, e.g., miR-146a and miR-146b. Finally, miRNAs can originate from different arms due to the precursor hairpin structure. Based on the orientation of the miRNA along the complementary sequence, miRNAs from different termini are differentiated by the suffix 3-p or 5-p, e.g., miR-192-5p and miR-192-3p [28].

The regulation abilities of miRNAs led to a surge in research activities. Differentially expressed miRNAs and their contribution to changed cell function were investigated in pathological conditions, e.g., cancer [29], autoimmune diseases [30], or trauma [31,32]. In peripheral nerves, certain miRNAs have an impact on crucial processes such as extracellular matrix remodeling, axonal growth, intracellular signaling, proliferation, migration, cell adhesion and neurogenesis by regulating specific mRNAs, and proteins [33,34,35]. Thereby, miRNAs contribute as regulators and modulators to the manifold signaling pathways activated and silenced during nerve regeneration. Shedding light on the function of single miRNAs might help us to understand the various processes during a peripheral nerve injury and regeneration and to identify novel targets for therapeutic use. This review summarizes the impact of miRNAs, which play a key role in nerve regeneration, and highlights their specific temporal expression profile during peripheral nerve injuries.

## 2. Materials and Methods

This systemic literature review was performed in accordance with recent PRISMA guidelines [36].

### 2.1. Inclusion Criteria

All studies which investigated miRNA in acute peripheral nerve injury models in rodents were taken into consideration. Two general types of articles could be identified and were included:(a)articles investigating a single miRNA, its targets and function in SC cultures;(b)publications exploring overall expression of miRNAs in acute/traumatic peripheral nerve lesion models (crush or transection) with at least with a microarray or full sequencing approach.

### 2.2. Exclusion Criteria

Publications with inadequate or incomprehensible methods were excluded from this review. For the sake of clarity, miRNAs with the nomenclature “miR-sc” were excluded from analysis.

### 2.3. Search Algorithm Results

Systemic literature research was performed with the searching engines PubMed and Google Scholar. For the literature search, we established a search algorithm with the key words “miRNA or microRNA or miR” combined with the terms “nerve injury”, “nerve crush”, “nerve lesion”, “Schwann cell”, “nerve regeneration”, “axotomy”, and “nerve transection”. The results from systematic literature research with the selection process are summarized by the flow diagram below (Figure 1).

### 2.4. Data Extraction

Two authors independently extracted relevant data. First, single miRNAs and their function were identified, which were analyzed in Schwann cell cultures. Subsequently, relevant data were extracted from records, including information about these miRNAs from microarray or full sequencing approaches. Due to different methods used by the groups, only qualitative conclusions were taken from the results (up-, downregulation, or no changes). For creating the respective expression profiles, results were sorted by the timepoints and summarized. In case of deviant results from different groups, the majority was taken to define either an up- or downregulation. miRNAs with an equal distribution of results were stated as not clear.

## 3. Results

A total number of 929 papers could be identified by the search strategy in accordance with PRISMA guidelines [36]. From these, 277 hits were identified as duplicates and excluded from further analysis. Upon closer examination of the titles and abstracts, 583 further papers could be eliminated for not meeting the inclusion criteria, leaving 69 records. In the full-text analysis, two papers had to be removed due to concerns about the result reproducibility and 31 based on missing inclusion criteria. A total number of 36 papers were included into the analysis. Of those, 28 papers exclusively investigated specific miRNAs, their function, and targets in SCs (Table 1).

Eight papers analyzed the expression profiles upon nerve lesion (Table 2). Three of them additionally studied a single miRNA. The results of these groups were used to create a schematic expression profile for the 28 identified miRNAs.

### 3.1. Expression Profiles of miRNA upon Peripheral Nerve Injuries

Out of the eight groups analyzing expression profiles, four research groups used a mouse and a rat model, respectively. Five publications included exclusively male animals [34,45,53,66,67], Adilakshmi et al. (2012) [65] used female mice only, and Viader et al. (2011) [58] included both genders. For one publication, no information about the gender could be found [64]. All groups studied an acute sciatic nerve lesion model, with six causing a lesion by transection and two by crush. Concerning the time frame, Adilakshmi et al. (2012) examined short-term miRNA expression in the first 24 h after injury, and six groups focused on the acute to subacute settings (1 day to 14 days). Only Sullivan et al. (2018) provided results in a long-term setting up to 90 days after nerve transection [53]. By comparing miRNA expression between mice and rats, most miRNAs were up- or downregulated concurrently, with more discrepancies found between studies using the rat model. These findings are in accordance with reports of high conservation among different species [68].

Three characteristic miRNA expression profiles—continuous upregulation, downregulation, or initial downregulation followed by an upregulation—were identified (Table 3). First, miRNAs were found to be upregulated within the first two weeks such as miR-21, miR-29a, miR-29b, miR-132, miR-146b, miR-221, miR-222, miR-485, miR-3075, and miR-3099. Some of those were found by Sullivan et al. (2018) to be upregulated throughout the whole regeneration process for up to 90 days [53]. Second, miR-30, miR-129, miR-140, miR-340, and members of the let-7 family were found to be initially downregulated during the first week after nerve lesion with a higher expression from day 14 onward. Representatives of the third profile, miR-1, miR-9, miR-34, miR-124, miR-138, and miR-192, showed a mirror-inverted behavior to the first profile, being downregulated upon nerve lesion for up to 90 days.

### 3.2. Single miRNAs Influence Certain Nerve Regeneration Incrementally

As described at a previous point in this review, SCs undergo a transdifferentiation process in response to a nerve lesion [8,9]. The repair phenotype is characterized by an increased proliferation and migration driven by activator protein 1 (AP-1) transcription factor and by a downregulation of myelination program through decreased levels of early growth response 2 (EGR2/KROX20) [34]. Further, inflammation processes play an important role [9]. Certain miRNAs show characteristic expression profiles contributing to the changed behavior of SCs (Figure 2).

#### 3.2.1. Inflammation

The initial response to Wallerian degeneration is accompanied by inflammation processes [9]. This phase is characterized by increased migration and proliferation of SCs with phagocytosis of debris and myelin clearance in cooperation with resident and invading macrophages [8].

##### miR-182

In nerve lesion models, fibroblast growth factor 9 (FGF9) has proinflammatory effects and furthermore inhibits myelination [69]. Deficiency of FGF9 results in a deteriorated infiltration of macrophages and consequently delayed debris clearance. Moreover, decreased expression of the cytokines monocyte chemotactic protein 1 (Mcp1), tumor necrosis factor a (Tnfa), and interleukin 1b (Il-1b) was detected due to the absence of FGF9 [70]. Neurotrimin (NTM), a member of the neural cell adhesion molecules (NCAM) family, promotes SC migration and plays a crucial role in axon guidance during nerve regeneration [71]. miR-182 has been reported to transcriptionally and translationally decrease levels of FGF9 and NTM [38]. In vivo, miR-182 is downregulated throughout the nerve regeneration process from the fourth day on [45,53]. Thus, decreased levels of miR-182 allow crucial inflammatory processes to occur with increased migration during the initial phase of an acute nerve lesion.

##### miR-340

Tissue plasminogen activator (TPA) is essential for debris removal in the initial phase by cleaving cell adhesion proteins as well as cell and extracellular matrix fragments resulting from Wallerian degeneration. Li et al. (2017) showed that miR-340 targets TPA, leading to a reduced SC migration and debris removal [37]. After nerve injury, miR-340 is found to be downregulated over the first week, and TPA is correspondingly overexpressed, with a peak in expression on days four to seven. In the concluding period of nerve regeneration 30 days after the nerve lesion, miR-340 is upregulated again, which could be indicative for a reregulation of TPA back to physiological levels.

#### 3.2.2. Proliferation

Groups analyzing mRNA expression profiles upon nerve lesion found significant peaks of proteins responsible for an increased proliferation, migration of SCs and impaired myelination around the 4th to 14th day [58,72]. The major repair genes c-Jun, nerve growth factor (Ngf), brain-derived neurotrophic factor (Bdnf), glial cell-derived neurotrophic factor (Gdnf), and SRY-related HMG-box 2 (Sox2) were found to be upregulated from the first day on [8,34,44,73,74].

##### miR-1

BDNF promotes axon elongation and inhibits apoptosis in neuronal cells [75]. It has been found that miR-1 targets the 3′-UTR binding side of Bdnf, resulting in a degradation of this regulatory factor [39]. In this context, the peak level of Bdnf around day seven correlates well with the nadir of miR-1 attained during the same period [39,45]. Beyond BNDF, miR-1 controls N-myc downstream-regulated gene 3 (NDRG3), a promoter of migration and proliferation, by direct degradation and translation suppression [40,76]. Furthermore, the application of a miR-1b-mimic results in decreased levels of Krüppel-like-factor 7 (KLF7), ciliary neurotrophic factor (CNTF), and NGF by increasing levels of the proapoptotic C-caspase 3 [41]. Accordingly, miR-1 expression is continuously downregulated during the proliferative phase and does not reach its base levels until day 90.

##### miR-21 and miR-124

NGF is known to be regulated by the transcription factor signal transducer and activator of transcription 3 (STAT3) [77]. STAT3 is regulated by miR-124, and a downregulation of miR-124 for up to 90 days after nerve lesion leads to increased levels of STAT3 and subsequently induces the transcription of NGF [42,78]. NGF itself promotes the gene expression of miR-21, a pro-proliferative miRNA.

Exemplary for miR-21-regulated molecules, Ning et al. (2020) found lowered expression of transforming growth factor b1 (Tgf b1) and ephrin type-A receptor 4 (Epha 4) as well as caspase family apoptosis proteins in dorsal root ganglion (DRG) and SC cultures following the application of miR-21 mimics [43]. In addition, miR-21 targets the 3′-UTR region of Sprouty homolog 2 (Spry2), the phosphatase and tensin homolog (Pten), and the tissue inhibitor of metalloproteinase 3 (Timp3). All three are essential tumor suppressors promoting programmed apoptosis [79,80,81]. In particular, SPRY2 and PTEN act as inhibitors of the Ras/Raf/ERK and Pi3k/AKT/mTOR pathways [82]. Furthermore, PTEN is involved in the regulation of NGF. PTEN directly dephosphorylates the transcription factor cAMP response element-binding protein (CREB), which leads to a suppression of NGF expression [83]. In combination with miR-222, miR-21 was shown to decrease TIMP3 levels in DRGs [84]. In line with increased proliferation of SCs during nerve regeneration, miR-21 expression is consistently upregulated for up to 90 days.

##### miR-221/222

The miR-221/222 complex is an additional participant in the regulation network of the cell cycle. Zhou et al. (2012) could demonstrate the degradation of PTEN by miR-222 in vitro and in vivo: the administration of miR-222 mimics caused a drop in PTEN levels [29] and an increase in phospho-mTOR levels [85]. Furthermore, the miR-221/miR-222 complex targets longevity assurance homolog 2 of yeast LAG (Lass2) by directly binding to the 3′-UTR region [45]. Like PTEN and TIMP3, LASS2 belongs to the tumor suppressors or, specifically, metastasis suppressors [86]. Thus, miR-221 and miR-222 attenuate the inhibition of SC migration and proliferation evoked by LASS2 [45]. In addition, miR-221 might contribute to the termination of the myelination program by targeting NGF1-A binding protein 1 (NAB1), the occurrence of which is indispensable for the regulation of Egr2 [46,87]. Consistent with these findings, miR-221 and miR-222 are upregulated from an early phase after nerve lesion. This upregulation may be due to the increased release of BDNF by SCs upon nerve lesion [85], though the miR-221/222-complex remains upregulated for up to 90 days, even when remyelination already proceeds [53].

##### miR-192-5p

X-linked inhibitor of apoptosis protein (XIAP) is an inhibitor of apoptosis processes in cells with multiple targets involved in promoting apoptosis [88]. miR-192-5p targets XIAP, thereby promoting apoptosis of SCs. Consequently, the inhibition of miR-192-5p in vitro leads to a decrease of apoptosis-related proteins from the B-cell lymphoma proteins (BCL-2) and an increase in neurotrophic factors [47]. The initial boost of proliferation could also be caused by the downregulation of miR-192-5p, which was detected over the first two weeks.

##### miR-3099

Another regulator of SC proliferation might be miR-3099 [48]. Increased levels of miR-3099 have been found upon nerve injury, and treatment of SCs with miR-3099 leads to an increased proliferation and migration in vitro [48]. This is consistent with an upregulation of miR-3099 for up to two weeks. Furthermore, miR-3099 has been shown to play a crucial role in stimulating neurogenesis during embryonic neural development [89]. However, an analysis of the precise targets of miR-3099 is, so far, missing, and further experiments are required to fully define the role of miR-3099 during nerve regeneration.

##### miR-146b

KLF7 stimulates nerve regeneration by promoting axonal outgrowth and neural cell proliferation and survival [90,91]. Li et al. (2018) investigated the correlation of KLF7 mRNA and protein levels and miR-146b expression upon sciatic nerve transection [49]. The results revealed an inverse correlation between miR-146b and KLF7. A transfection with miR-146b packed in lentivirus vectors decreased the appearance of KLF7 mRNA and protein in vitro, resulting in inhibition of neurite outgrowth. Moreover, the local administration of anti-miR-146b upon nerve lesion in vivo yielded higher levels of KLF7, NGF, and tyrosine receptor kinase (TRK) A and B. Accordingly, the inhibition of miR-146b resulted in improved functional recovery with increased myelination, muscle motor endplate regeneration, and recovery in gait analysis [49]. Li et al. (2018) noted significantly decreased levels of miR-146b one week after nerve lesion [49], whereas the expression profiles identified in this review showed a continuously upregulation of miR-146b throughout the period of up to 90 days postinjury.

##### miR-210

miR-210 increases proliferation and migration in vitro by inhibiting apoptosis [50]. One possible underlying pathway could be the degradation of ephrin-A3 (EFNA3), which was found in decreased levels upon the administration of miR-210 in vitro [92]. These observations are in concordance with Wang et al. (2014). They detected an upregulation of miR-210 in malignant peripheral nerve sheath tumors (MPNST) going with a downregulation of EFNA4 and augmented proliferation and colony formation and migration [93]. There might also be a correlation between miR-210 and the myelination process; Zhang et al. (2017) described increased levels of myelin basic protein (MBP) and growth-associated protein (GAP-43) but decreased levels of myelin-associated glycoprotein (MAG) in vitro after application of miR-210 mimics to SCs [50]. Since miR-210 is found to be downregulated in the first week and only upregulated two weeks after nerve injury, the expression profiles would correlate more with the expected effects on myelination than on proliferation. 

#### 3.2.3. Migration

Amplified migration of cells is characteristic shortly after nerve lesion. Collagen triple helix repeat containing 1 (CTHRC1) embedded in the Wingless and Int-1/Planar cell polarity (WNT–PCP) pathway [51], Netrin-1 (NTN1), and deleted in colorectal carcinoma (DCC) provides for SC migration [52].

##### miR-9

CTHRC1 and DCC are regulated by miR-9. miR-9 binds directly to the 3′-UTR, degrading DCC, which impedes SCs migration [52]. Furthermore, the transfection with miR-9 abolishes CTHRC1, resulting in diminished SC migration in vitro and in vivo [93]. miR-9 was found to be downregulated over the whole investigated period of 90 days, which is in conformity with the higher mRNA and protein levels of DCC found after nerve lesion [52].

##### miR-138-5p

miR-138-5p is a further modulator of SCs migration. It negatively affects the migration rate of SCs by degrading vimentin, one of the main proteins of the SC cytoskeleton [53]. The expression of miR-138-5p is regulated by EGR2 and SOX10, which will be described in a later chapter [94]. Corresponding to the initial silencing of Egr2, miR-138-5p is found to be downregulated from day one onward. Despite rising levels of EGR2, the decreased expression of miR-138-5p remains for 90 days. This implies additional regulatory mechanisms of miR-138-5p that have not yet been identified.

##### miR-129

Insulin-like growth factor 1 (IGF-1), one of the activators of the PI3K/AKT pathway, promotes neurite outgrowth, SC proliferation, and migration [95] and shows therapeutic effects in nerve regeneration [96]. miR-129 targets IGF-1 and the use of miR-129 inhibitor is associated with augmented neurite outgrowth and SC migration due to the attenuated inhibition of the PI3K/AKT pathway [54]. miR-129 expression can be characterized by the second expression profile with an initial downregulation followed by a subsequent upregulation two weeks after nerve lesion.

##### miR-34a, miR-132, and miR-3075

5′-AMP-activated protein kinase subunit gamma 3 (PRKAG3) is a protein inhibiting cell migration. PRKAG3 was identified as a target of miR-132 [55]. miR-34a directly targets contactin 2 (CNTN2), which is a cell adhesion protein, and positively correlates with SC migration. Moreover, the knockdown of miR-34a increased axonal outgrowth and proliferation of SCs [57]. Wang et al. (2020) discovered another regulator of CNTN 2 in miR-3075. In vitro, the negative regulation of Cntn2 by miR-3075 only affected SC migration, but not proliferation [56]. miR-34a was found to be downregulated, while miR-132 was upregulated in the first two weeks after nerve injury, which is well consistent with their predicted functions. On the contrary, miR-3075 is found to be upregulated during the acute phase. This expression profile would be detrimental for migration during primary phase of Wallerian degeneration and nerve regeneration. It leads to the assumption that miR-3075 might play only a minor role by slightly counter-regulating migration.

#### 3.2.4. Myelination

Myelination is driven by EGR2, which is induced by neuregulin 1 (NRG1) and by SOX10 [34]. The most characteristic compounds of the myelin sheath positively affected by EGR2 are the MBP, peripheral myelin protein 22 (PMP22), and myelin protein zero (MPZ) [97]. During the initial phase after an acute nerve lesion, SCs cease myelination and transdifferentiate into repair program by silencing Egr2 [34]. 

##### miR-34a and let-7 family 

NOTCH1 is a negative regulator of myelination and Egr2 expression by simultaneously inducing proliferation of SCs [97]. The aforementioned miR-34a directly targets Notch1 and Cyclin D1 (Ccdn1) [58]. In addition, Notch1 is negatively regulated by the Lin28/let-7 axis [59]. The downregulation of miR-34a and members of the let-7 family in the first week after nerve injury leads to an upregulation of Notch1, resulting in suppressed myelination. Moreover, in vivo and in vitro experiments have demonstrated that members of the let-7 family regulate NGF at the posttranscriptional level, which is essential for proliferation of the dedifferentiated SCs [60]. The upregulation of the let-7 family from day 14 to 90 after nerve lesion indicates the downregulation of Notch1 and thus higher EGR2 levels, inducing remyelination of outgrowing axons.

##### miR-140

Further, Egr2 is directly targeted by miR-140, leading to a reduction of the myelin compounds [58]. However, in our analysis of previously published expression profiles, miR-140 was found to be initially downregulated and then upregulated over the next three months as myelination occurs. This expression profile is completely different from the EGR2 levels, which initially decrease with a subsequent upregulation in the subacute period of nerve regeneration [98].

##### miR-29a and miR-29b 

Verrier et al. (2009) investigated certain miRNAs targeting PMP22 in SC cultures. An inverse correlation between members of the miR-29 family and PMP22 levels was demonstrated by transfecting SC with miR-29 mimics and inhibitors [61]. Thus, the expression profile of upregulated miR-29a and miR-29b over the first two weeks matches the profiles of miRNAs inhibiting myelination.

##### miR-485-5p

Knockout models showed the essential role of cell division control protein 42 (CDC42) [99] and Ras-related C3 botulinum toxin substrate1 (RAC1) [100] for a successful myelination acting on the Neurofibromatosis Type 2 (NF2)/merlin pathway. CDC42 and RAC1 are regulated by miR-485-5p. Moreover, the application of miR-485-5p mimics resulted in an impaired myelination with reduced levels of EGR2, and MBP in vitro [62]. The upregulation of miR-485-5p detected in the acute phase may contribute to the arrested myelination program.

##### miR-30c

A promyelinating effect was shown after the application of miR-30c agomirs in vivo and in vitro by Yi et al. (2017). They detected increased Mbp expression in vitro in SC and DRG coculture and subsequently enhanced remyelination of lesioned nerves in vivo. Nevertheless, they failed to identify a certain target of miR-30c. miR-30c following nerve injury was downregulated in this setting, with the lowest levels reached on day four by slowly returning to preinjury levels over the course of 28 days [63]. The expression profiles can be confirmed by the results obtained in this review, with an initial downregulation and subsequent upregulation at the second week. Together with the expression profiles of other myelination-regulating miRNAs, the results suggest an onset of remyelination around the second week after an injury.

## 4. Discussion

Several limitations of this systemic review exist. Due to the limited number of publications investigating expression profiles and a heterogeneity regarding species, gender, lesion type, method, lesion type, and the lack of standardized tissue collection, the expression profiles were only analyzed on qualitative parameters. Thus, the definite interpretation is restricted to a schematic overview. The effects described for certain miRNAs cannot be quantified due to aforementioned limitations. Consequently, we waived the comparison of single miRNAs in this review. Moreover, most correlations between single miRNAs and their targets were investigated by one publication each. Hence, the reproducibility of the findings remains questionable. The lack of coherence to nomenclature resulted in the exclusion of miRNAs termed “miR-sc” in this review. 

In this review, we summarized miRNAs associated with nerve regeneration and their impact on related processes of Schwann cells. By a systemic approach, 28 miRNAs could be identified in this review, which were investigated in vitro in SCs or in vivo. Results from full sequencing approaches identified more than 200 miRNAs, which were differently expressed [66]. The main proportion of miRNAs participating in nerve injuries and regeneration remains unexplored.

### 4.1. Expression Profiles and Function

miRNA expression profiles were included from experiments which analyzed peripheral nerve tissue either from the proximal or distal site of the lesion. It is noteworthy that, in most experiments, whole sciatic nerve tissue with SC, macrophages, fibroblast, and other resident cell types was investigated. Hence, an assignment of cell-specific miRNAs is not given by most of the mentioned publications [58,65]. A further, single-cell-based investigation is needed to identify cell-specific miRNAs and their roles in the regeneration process.

The investigation of miRNAs sheds light on the highly complex mechanisms of nerve regeneration with a plethora of participating molecules and processes. For most miRNAs identified in this review, the expression profiles matched the known expression profiles of mRNAs. The arrested myelination program and SC dedifferentiation with increased migration and proliferation around the fourth day fit with the patterns of miRNAs regulating proliferation, migration, and myelination.

### 4.2. Controversial miRNAs

For miR-140, miR-146b, miR-210, and miR-3075, the predicted effects of protein regulation were contrary to the expression profiles identified. miR-140 suppresses Egr2 directly and therefore impairs the myelination program [58]. This miRNA is initially downregulated, and its levels rise again from day 14 onward. In the same period, Egr2 levels are decreased by a subsequent upregulation [34]. Hence, miR-140 may not be the key regulator of Egr2 expression and is more likely play an important role as a counterweight to miR-34a [58]. The role of miR-146b might be controversial as an upregulation acts inhibitory on proliferation by decreasing levels of KLF7 [49]. Same as miR-3075, which is upregulated throughout the acute phase after an injury and impedes migration [56]. miR-210 is a proliferation- and migration-promoting miRNA [50,92]. Nonetheless, miR-210 is downregulated during the first two weeks following nerve lesion. Upcoming results will have to show whether results for controversial miRNAs and their targets were correctly identified or if these miRNAs may be embedded in a more complex network with counter-modulation effects. As previously mentioned, the prior studies were not able to differentiate the miRNAs by their origin type of cell. A critical look needs to be taken at the miRNA expression profiles at a single cell resolution, as the elevated levels of conflicting miRNAs could be allocated to auxiliary cell types, such as macrophages or fibroblasts rather than SCs.

### 4.3. miRNAs in Therapeutic Approaches

Knowing the potential of miRNAs to influence cellular processes, it is evident that certain miRNAs could be therapeutically administered to enhance nerve regeneration. Several groups demonstrated an improvement of functional outcome in nerve lesion models after the application of miRNA mimics or miRNA inhibitors [33,49,60,92]. Besides the use in acute nerve injuries, the field of application is widespread due to the possibility of a two-way modulation of certain processes. In particular, an enhanced inflammation is beneficial during Wallerian degeneration [70]. The inhibition of miR-182 leads to higher levels of the proinflammatory FGF9 and thus a facilitated invasion of immune cells and more rapid clearance of debris [38]. On the contrary, in chronic demyelination diseases, inflammation and Wallerian degeneration are detrimental. Cai et al. (2018) revealed the therapeutic potential of miR-182 in a diabetic mice model. The administration of miR-182 reversed the dysregulation of the sodium channel Nav 1.7, lowered FGF9, and thus resulted in an amelioration of pain levels [101].

The existence of PTEN and SPRY 2 is crucial for the regulation of axonal outgrowth and SC migration and proliferation in homeostasis. During the acute phase of nerve injury, high levels of PTEN and SRPY2 are unfavorable [82,102]. Consequently, a reduction of these two proliferation-inhibitory proteins by the application of miR-21 or miR-221/222 is necessary to accelerate nerve regeneration. However, highly proliferative tumor cells are using the same mechanisms to evade the growth control. miR-21 was found to be dysregulated in Schwann cells, and the underlying lack of PTEN and SPRY 2 resulted in uncontrolled tumor growth [103,104]. Analogously, miR-210 is upregulated in malignant peripheral nerve sheath tumors [93]. Consequentially, the application or inhibition of the same proliferation-promoting miRNAs might be indicated after a nerve lesion and in anticancer therapy, respectively.

In summary, the application of selected miRNAs demonstrated therapeutic effects and resulted in an enhanced nerve regeneration in nerve lesion models or an alleviated inflammation in diabetic neuropathies. However, at this point, no prediction about the long-term effects and the potency of miRNAs is possible. Furthermore, forthcoming research will have to address the half-life and the impact of miRNAs in vitro and in vivo.

## 5. Conclusions

miRNAs play a tremendous role as modulators of cellular function during peripheral nerve injuries and regeneration. Through the modulation of key regulatory molecules, these RNAs are involved in a plethora of processes such as inflammation, myelination, proliferation, and migration. A total number of 28 miRNAs could be identified and characterized at the present status of research. However, most miRNAs that are differentially expressed as a reaction to nerve lesions are still not investigated, and little is known about their targets and their roles. The possibility of either applying inhibitors or mimics of a selected miRNA enables researchers to alter inflammation, proliferation, myelination, or migration in a two-sided way. That provides us with a therapeutic opportunity to promote desired or readjust dysregulated processes, respectively, in a targeted manner. Their potential in precision medicine warrants further research on the field of miRNAs.

## Figures and Tables

**Figure 1 ijms-23-03440-f001:**
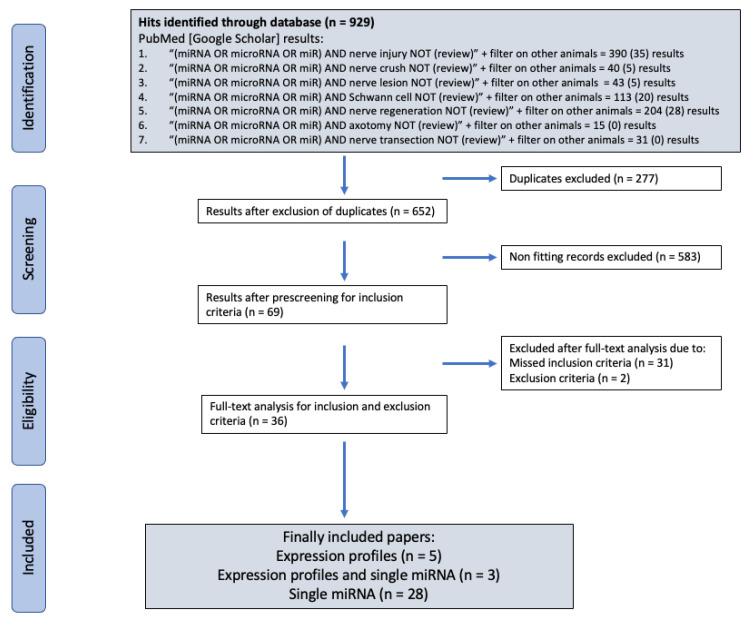
Overview search algorithm and results.

**Figure 2 ijms-23-03440-f002:**
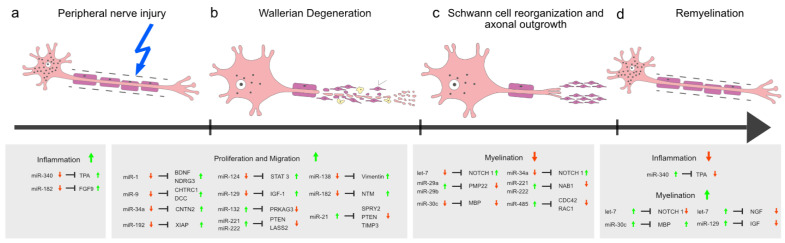
Schematic overview of processes following peripheral nerve injury and accompanying miRNA expression patterns. Following peripheral nerve injury (**a**), Wallerian degeneration (**b**) occurs with a transdifferentiation of Schwann cells and inflammation. Subsequentially, Schwann cells align and start with building Bands of Büngner, supporting the axonal outgrowth through the lesion site (**c**). Finally, Schwann cells remyelinate the fresh-grown axons returning in their mature cell type (**d**). miRNAs and their respective targets subdivided in “inflammation”, “proliferation and migration”, and “myelination” overlap temporally in the stages “Wallerian Degeneration” (**b**) and “Schwann cell reorganization and axon outgrowth” (**c**).

**Table 1 ijms-23-03440-t001:** List of identified miRNAs sorted by function.

miR	Function	Effect	Target	Author, Year, Reference
miR-340	Inflammation	Impair	TPA	Li et al. (2017), [37]
miR-182	Inflammation, Migration	Impair	FGF9, NTM	Yu et al. (2012), [38]
miR-1	Proliferation	Impair	BDNF	Yi et al. (2016), [39]
miR-1b	Proliferation, Migration	Impair	NDRG3	Liu et al. (2018), [40]
miR-1b	Proliferation	Impair	KLF7	Li et al. (2021), [41]
miR-124	Proliferation, Migration	Impair	STAT 3	Nagata et al. (2014), [42]
miR-21	Proliferation	Promote	TGFb1, EPHA4, TIMP3	Ning et al. (2020), [43]
miR-21	Proliferation	Promote	PTEN	Lopez-Leal et al. (2020), [44]
miR-221/222	Proliferation	Promote	LASS2	Yu et al. (2012), [45]
miR-221	Myelination	Impair	NAB1	Zhao et al. (2018), [46]
miR-192	Proliferation	Impair	XIAP	Liu et al. (2020), [47]
miR-3099	Proliferation, Migration	Promote	None identified	Liu et al. (2019), [48]
miR-146b	Proliferation	Impair	KLF7	Li et al. (2018), [49]
miR-210	Proliferation, Migration, Myelination	Promote proliferation and migration, impair myelination	None identified	Zhang et al. (2017), [50]
miR-9	Migration	Impair	CTHRC1	Zhou et al. (2014), [51]
miR-9/let-7 family	Migration	Impair	NTN1, DCC	Wang et al. (2019), [52]
miR-138	Migration	Impair	Vimentin	Sullivan et al. (2018), [53]
miR-129	Migration	Impair	IGF9	Zhu et al. (2018), [54]
miR-132	Migration	Promote	PRKAG3	Yao et al. (2016), [55]
miR-3075	Migration	Impair	CNTN2	Wang et al. (2018), [56]
miR-34a	Migration, Proliferation	Impair	CNTN2	Zou et al. (2020), [57]
miR-34a	Myelination, Proliferation	Promote myelination, Impair proliferation	NOTCH1, CCND1	Viader et al. (2011), [58]
miR-140	Myelination	Impair	EGR2	Viader et al. (2011), [58]
Lin28/let-7-axis	Myelination	Promote	NOTCH1	Gökbuget et al. (2018), [59]
let-7	Proliferation	Impair	NGF	Li et al. (2015), [60]
miR-29a/miR-29b	Myelination	Impair	PMP22	Verrier et al. (2009), [61]
miR-485-5p	Myelination	Impair	CDC42, RAC1	Zhang et al. (2020), [62]
miR-30c	Myelination	Impair	MBP	Yi et al. (2017), [63]

**Table 2 ijms-23-03440-t002:** Results for miRNA-expression profiles of miRNAs upon nerve lesion.

Author, Year, Reference	Method	Species	Gender	Lesion Model	Time Points (Postinjury)
Viader et al. (2011), [58]	Microarray	Mice	Both	Crush	4 d, 14 d
Wu et al. (2011), [64]	Microarray	Mice	Nd	Crush	4 d, 7 d
Adilakshimi et al. (2012), [65]	Microarray	Mice	Female	Cut	6 h, 24 h
Yu et al. (2012), [45]	Microarray	Rat	Male	Cut	1 d, 4 d, 7 d, 14 d
Sullivan et al. (2018), [53]	Microarray	Rat	Male	Cut	30 d, 60 d, 70 d
Arthur-Farraj et al. (2017), [34]	Illumina sequencing	Mice	Male	Cut	3 d, 7 d
Yu et al. (2011), [66]	Solexa sequencing	Rat	Male	Cut	1 d, 4 d, 7 d, 14 d
Li et al. (2011), [67]	Solexa sequencing	Rat	Male	Cut	1 d, 4 d, 7 d, 14 d

h = hours; d = days; Nd = not defined.

**Table 3 ijms-23-03440-t003:** miRNA-expression profiles generated from literature search.

	Acute Phase					Late		
miR	6 h	1 d	4 d	7 d	14 d	30 d	60 d	90 d
182		UP	DOWN	DOWN	NOT CLEAR	DOWN	DOWN	DOWN
340		DOWN	DOWN	DOWN	NOT CLEAR	UP	UP	UP
132		UP	UP	UP	UP			
3075		UP	UP	UP	UP			
9		DOWN	DOWN	DOWN	DOWN	DOWN	DOWN	DOWN
129		NOT CLEAR	DOWN	DOWN	DOWN	UP	UP	UP
138	DOWN	DOWN	DOWN	DOWN	DOWN	DOWN	DOWN	DOWN
29a		NOT CLEAR	UP	UP	UP			
29b		UP	UP	UP	UP			
485		UP	UP	NOT CLEAR	NOT CLEAR			
30c	DOWN	DOWN	DOWN	DOWN	UP			
140	DOWN	DOWN	DOWN	NOT CLEAR	UP	UP	UP	NORMAL
34a	DOWN	DOWN	DOWN	DOWN	NOT CLEAR			
let-7-a		DOWN	DOWN	DOWN	UP	NORMAL	UP	UP
let-7-b		DOWN	DOWN	DOWN	UP	NORMAL	NORMAL	UP
let-7-c		NORMAL	DOWN	DOWN	UP			
let-7-d		UP	DOWN	DOWN	UP	UP	UP	UP
let-7-e		UP	DOWN	DOWN	UP	NORMAL	UP	UP
let-7-f		NOT CLEAR	DOWN	DOWN	NORMAL	DOWN	NORMAL	UP
1		DOWN	DOWN	DOWN	DOWN	DOWN	DOWN	NORMAL
124		NOT CLEAR	DOWN	DOWN	DOWN	DOWN	DOWN	DOWN
192		DOWN	DOWN	DOWN	NOT CLEAR			
210		DOWN	DOWN	DOWN	UP			
146b	UP	UP	UP	UP	NOT CLEAR	UP	UP	UP
21		UP	UP	UP	UP	UP	UP	UP
221		UP	UP	UP	UP	UP	UP	UP
222		UP	DOWN	UP	UP	UP	UP	UP
3099		UP	UP	NOT CLEAR	UP			

In case of deviant literature conclusions, the results of the majority were taken. Results with no majority were defined as “NOT CLEAR” (in white). Upregulated miRNAs were marked green (“UP”); downregulated miRNAs red (“DOWN”); miRNA where no difference was found compared to healthy condition were defined as “NORMAL” (blue). If no data could be found for the corresponding time point and miRNA, the result was left blank; h = hours; d = days.

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
