# Peer review of "How miRNAs Regulate Schwann Cells during Peripheral Nerve Regeneration—A Systemic Review"

_ijms, 2022, doi:10.3390/ijms23073440_

Round 1
Reviewer 1 Report
The study describes how miRNAs regulate Schwann cells during peripheral nerve regeneration. This is a very interesting topic to search new treatment possibilities and to prepare new rehabilitation standards. Please add all limitations of the study. Please rewrite references according to MDPI rules.
Author Response
Dear reviewer, thank you for your constructive feedback.
Point 1: Please add all limitations of the study
Response 1: A section about the limitations was added to the discussion in the line 409-418.
Point 2: Please rewrite references according to MDPI rules.
Response 2: The references were adapted to MDPI rules with the Zotero "MDPI"template.
Reviewer 2 Report
The manuscript by Borger and co-authors describes the role of miRNA in Schwann cells during the regeneration of the peripheral nerve.
The manuscript is interesting. The authors performed a deep investigation of the literature correlating the miRNA expression involved in the molecular events necessary for the regeneration of the peripheral nerve.
The review is organized as a research article. This format is unusual, but the manuscript has a logical structure.
I have some suggestions that may improve the work.
-The authors have to include a figure showing the Wallerian Degeneration. Alternatively, they can move the figure 2a,b,c,d in section 1.1.
-The authors have to improve the introduction about miRNA. This section is too concise and confusing.
-The authors have to add a scheme reporting the molecular correlation of all miRNA listed in section 3.2
Author Response
Dear reviewer, thank you for your very constructive feedback.
Point 1: The authors have to include a figure showing the Wallerian Degeneration. Alternatively, they can move the figure 2a,b,c,d in section 1.1.
Response 1: We understand your objection and as you mentioned, figure 2a-d contains the Wallerian Degeneration. Further, it contains the differentially expressed miRNAs with their targets, which is part of the results section. Thus, we think it fits better in section 3, and a second figure about the Wallerian Degeneration would be superfluous to our opinion. Nevertheless, we trust your experience and would not hesitate to adopt the placement of the figure, if you say it is indispensable.
Point 2: The authors have to improve the introduction about miRNA. This section is too concise and confusing.
Response 2: We have rewritten, and restructured the chapter about miRNAs. Please find the renewed part in line 60 to 93. We hope the new version fulfills your criticism.
Point 3: The authors have to add a scheme reporting the molecular correlation of all miRNA listed in section 3.2
Response 3: Molecular correlations between all miRNAs, and their targets from section 3.2 are displayed in table 1, and in a schematic way in figure 2. Therefor, figure 2 might have to be enlarged, and placed more prominently, if it is hard to read.
Round 2
Reviewer 2 Report
no comments